# Classroom-Based Physical Activity as a Means to Improve Self-Efficacy and Academic Achievement among Normal-Weight and Overweight Youth

**DOI:** 10.3390/nu15092061

**Published:** 2023-04-25

**Authors:** Francesca Latino, Francesco Tafuri, Emma Saraiello, Domenico Tafuri

**Affiliations:** 1Faculty of Human Sciences, Pegaso University, 80100 Napoli, Italy; 2Heracle Lab Research in Educational Neuroscience, Niccolò Cusano University, 00166 Roma, Italy; 3Department of Movement Sciences and Wellness, University of Napoli “Parthenope”, 80100 Napoli, Italy

**Keywords:** academic performance, weight status, exercise, scholastic self-efficacy, cognitive function, lifestyle

## Abstract

Although physical activity has positive physical and mental health outcomes, particularly among adolescents, a significant percentage of young people maintain a largely sedentary lifestyle. Considering that the youths spend the greater part of the day at school, this is considered an ideal setting to foster active and healthy living. Consequently, this study is intended to investigate the connection between physical activity, self-efficacy and academic achievement in normal-weight and overweight adolescents. In total, 100 students (aged 14–15) from a public high school placed in the south of Italy were enrolled. They participated either in a 12-week classroom-based physical activity break program performed during science classes (60′/2 days per week) in which a nutritional educational program was carried out or in regular science lessons (60′/2 days per week). At the beginning and end of the intervention programs, a set of standardized motor evaluation tests (standing long jump test, Harvard step test, push up, sit and reach test), the scholastic self-efficacy test and the Amos 8-15 were administered. As a result, a meaningful Time × Group interaction for the self-efficacy variable and Amos 8-15 was observed in the intervention group. Specifically, they reported significant improvement in study skills, motivational factors, concentration and self-efficacy, as well as a decrease in anxiety and BMI (*p* < 0.001). No significant change was observed in the control group. The conclusions of this research underpin the notion that classroom-based physical activity break is a successful approach for enhancing students’ psycho-physical well-being, as well as academic achievement.

## 1. Introduction

The scholastic readiness and well-being of the youth are major concerns in the 21st century [1]. The school is seen as a perfect environment for promoting an active and healthy lifestyle, in order to improve the youngers’ well-being. Hence, the school should focus on being tasked with preventing obesity as well as the need to provide a high education and academic achievement for all students [2]. In fact, children and adolescents spend the greater part of their time attending school. Nevertheless, it should be highlighted that traditionally schools are mainly focused on increasing literacy and numeracy proficiency to the detriment of physically active behaviors [3]. In this regard, thanks to the fact that children and adolescents are accessible in this setting, many studies tried to focus on childhood obesity by developing interventions that may be used within the educational system [4].

Obesity, an excessive body fat storage state, is caused mostly by improper lifestyle [5,6]. It is the result of both an unhealthy diet and a lack of physical activity [7]. As World Health Organization (WHO) claimed, in recent times, the percentage of obesity has globally more than doubled among children and tripled among adolescents [8]. Moreover, they are not following the 2020 WHO physical activity guidelines, which urges young people from 5 to 17 years to reach at least 60 min of moderate-to-vigorous physical activity (MVPA) per day [9].

The maintenance of a desirable level of physical activity plays a key role in the improvement of cognitive, motor, and social skills. It is widely acknowledged that decreased levels of physical activity represent a crucial risk factor for the dramatical rising levels of obesity [10]. Furthermore, fitness seems related to better cognitive function and academic performance, as well as to the own psycho-social behavior [11,12,13,14,15].

Several researchers suggest that physical activity can be associated with cognition over the course of development [16]. They argued that young people participating in an increased amount of physical activity better execute attentional tasks which involve a higher level of cognitive control, such as planning, organization, problem solving, working memory, motor control and inhibitory control [17,18].

To explain the positive influence of physical activity on academic performance, we made the assumption that the related neurophysiological changes in the brain are due to the increase in brain blood flow, better neuroelectric function, and solicitation of the brain-derived neurotrophic factor (BDNF) release. These are all factors that encourage learning and keep cognitive functions by enhancing synaptic plasticity [19,20].

Moreover, the combination between weight status, eating habits and academic achievement is affected by individuals’ psychosocial variables, such as self-efficacy [21,22]. Self-efficacy concerns the conviction that one can effectively participate in a certain behavior [23].

Perceived self-efficacy is the result of a self-referential and self-regulated system that guides and directs the behavior, orients the individual’s relationship with the environment and creates the conditions for the development of new experiences and abilities [24].

Individuals who have low levels of self-efficacy often present closing and avoidance behaviors and poor performance or failure, while people with a higher level of perceived self-efficacy have a good chance of obtaining satisfactory results. Thus, those who believe they will be able to accomplish a task will obtain higher performance than those who assess it negatively [25]. This behavior deeply affects both weight control and academic performance.

Concerning that, a growing body of literature has documented that fatness and academic failure seem to be related to poor self-efficacy [26,27,28,29] and that physical activity is able to solicit significant behavior changes [30,31]. Physical activity has the potential to boost self-efficacy beliefs that are associated with changes in the person’s mood when exercise ends [32,33]. This happens thanks to the fact that the experience of physical activity encourages the ability to leverage one’s mental and social attitude to gain an extensive repertory of motor skills which can be expressed in various environments and in all daily activities [34].

In light of this theoretical framework, it is possible to indicate that perceived self-efficacy improves thanks to physical activity and that physical activity plays a crucial role in mediating self-efficacy and the impact of the interventions on dietary behavior and academic performance. Consequently, an intervention that combines physical activity, nutritional education and academic achievement could provide an unrepeatable occasion to intervene and propose a program that both enhances health and academic performance. In the school context, an alternative way to practice physical activity is through an “active classroom”, where students incorporate movement into learning. It represents an opportunity to help them retain knowledge in a meaningful way, as well as to acquire an active lifestyle.

Several studies suggest that classroom-based PA performed for 10–15 min, 2 or 3 days per week, has a positive impact on increasing PA levels, classroom and learning behavior, as well as decreased weight status [35,36]. Concerning that, in a 2019 Italian study, Calella et al. [37] conducted a classroom-based active break (CABs) intervention that integrates PA during school time and assessed its potential effect on reducing inactivity and overweight in primary school children. As they argued, the CAB program is a safe tool to reduce inactivity and overweight and increase moderate/vigorous PA. According to Masini et al. [38], active breaks (CABs) led by teachers inside the classroom represent a good strategy for promoting PA. In their 2020 study, the authors supported the feasibility and effectiveness in terms of increased PA level of a CAB program in children attending primary school. Similarly, Di Maglie and colleagues [39] observed that the school-based intervention program represents an effective strategy for decreasing the number of children with obesity. It is also interesting to mention a study by Fiori et al. [40], in which the authors compared the quality rather than the duration of time spent in the gym during school hours. They indicated that quality PA was effective in improving children’s physical fitness, significantly decreasing their body weight.

Therefore, the purpose of this work was to analyze the link between physical activity, body mass index (BMI), self-efficacy and academic achievement in adolescents. Moreover, this work aimed to establish if there are meaningful differences in perceived self-efficacy and school performance between overweight and non-overweight high school students measured through standardized assessment tests. We assumed that a quantifiable disparity of results would be found among overweight and non-overweight students at the end of a classroom-based physical activity program.

## 2. Methods

### 2.1. Study Design

The research is regarded as a controlled study to analyze the mediating role of a school-based physical activity program on overweight adolescents’ self-efficacy and academic achievement. The research was carried out in a high school. The participants were involved in a scholastic intervention that combined lessons in nutritional education with physical activities carried out during science classes. Thus, the intervention program involved 24 lessons of moderate-to-vigorous aerobic (MVPA) exercises for the intervention group and a regular science lesson for the control group, during which no practice was carried out. The interventions consisted of 2 weekly lessons of sciences of the duration of 60 min in which a classroom-based physical activity lasting 20 min was integrated. The science classes consisted of nutritional educational lessons. The evaluations were performed before and at the end of the intervention programs (Figure 1).

### 2.2. Participants

The students enrolled were from a public high school placed in the south of Italy. One hundred students, as a convenient sample, were recruited with an age range of 14–15 years (M age = 14.64, *SD* = ±0.47). Participation in this trial was on a voluntary basis, and all the first- and second-year students were invited to participate. Of these students, 119 accepted the invite. To be included in this study, participants had to be between 14 and 15 years of age, be capable of finishing a moderate-to-vigorous intensity aerobic exercise session and attending the selected school, and be ready to avoid all physical activity beyond those indicated by the protocol. Exclusion criteria included acute illness at the time of testing, injuries that prevent performing exercises, and those unable to avoid physical activity beyond those indicated by the protocol. In total, 109 students met the inclusion criteria and were asked to participate in the trial. Of those recruited, 100 (56 males and 44 females) accepted to be involved in this research and rounded out the evaluations at baseline and post, while 9 of them refused because of personal issues.

As a result, the final sample consisted of 100 participants, of which 41 were overweight and 59 were non-overweight (41% overweight and 59% non-overweight), who completed the assessments at baseline and post-intervention. We matched participants randomly to one of two treatment conditions, and the percentage of overweight children was 46% for the experimental group (EG) and 36% for the control group (CG) (EG: overweight n = 23, non-overweight n = 27; CG: overweight n = 18, non-overweight n = 32).

The participants were sent an email to be advised in advance of their inclusion in the study. Following that, a second email was sent to the parents of all students, urging them to read a briefing that highlighted the goals of the trial and produce written consent for the participation of their kids in the research. Consent forms were signed by all parents prior to the start of the study. They contained the purpose of the study, the selection criteria, the procedures, the benefits and risks, alternatives, confidentiality, withdrawal and an injury disclaimer. Participants were assigned in a random way to one of two programs (EG n = 50; CG n = 50).

Using G*Power (version 3.1.9.6), an a priori power analysis was completed and showed that a sample size of 74 would provide a statistical power (α = 0.05, 1-β = 0.80) to identify a medium effect size (f = 0.25 or 0.4), given a coefficient of correlation *p* = 0.80 with 95% power and α = 0.05, using a within-between mixed design. To prevent experimental mortality due to participants leaving, 100 subjects were enrolled.

The anonymity of all participants was ensured by the researchers. The study was conducted from October 2022 to December 2022. The research was conducted based on the Declaration of Helsinki.

### 2.3. Procedures

Initially (one week before), all participants were invited to participate in a briefing especially conceived to provide information concerning the program. Moreover, at that time, the motivation of participants was verified.

Participants were assessed for anthropometric measurements, including BMI, in order to place each student into the non-overweight or overweight category. After the BMI assessment, the researchers assigned participants to one of the two program conditions in a random way. In addition, to establish the beginning level of each subject’s physical fitness, standardized motor evaluation tests were utilized. Moreover, two cognitive tests were administered to examine the student’s self-efficacy perception and academic performance.

The students were tested individually and undertook each test in the same order, at the same time of the day, and under similar trial conditions. They completed the evaluations two days before and immediately after the trial to enable the impact assessment of the physical activity program. All evaluations and both physical activity programs were outlined, supervised and carried out by 2 certified physical education teachers.

### 2.4. Measures

#### 2.4.1. Anthropometric Measures

A standard protocol was followed, and standardized instruments were used to carry out the anthropometric evaluations [35]. Height and weight measurements were performed in a room where the participant’s privacy was ensured. They carried out the measurements three times for each participant, and the verification was always performed by the same experienced researcher in order to avoid incurring errors. Participants were weighed using a portable medical floor scale with a precision of 0.1 kg (Charder’s MS6121R, Charder Electronic Co., Taichung City, Taiwan). A wall-mounted digital stadiometer was used to measure height to the nearest 0.1 cm (Charder’s HM200D, Charder Electronic Co., Taichung City, Taiwan). The height and weight measurements were used to calculate the body mass index (BMI) (kg/m^2^) [36]. BMI was converted to BMI percentile, and the international BMI cutoff points were utilized to classify overweight and non-overweight students [37]. BMI evaluation was performed before and after the intervention programs.

#### 2.4.2. Motor Tests

The measurements of physical fitness involved 4 evaluation tests:Standing long jump test, a measure used to evaluate lower-body horizontal explosiveness [38].Harvard Step test, utilized to evaluate aerobic fitness [39].Push-up test, employed to measure upper-body strength and endurance [40].Sit and reach, which measures the extensibility of the hamstring muscles and lower back [41].

These tests were chosen because they are easier to undertake, take little time and require simple equipment [42]. For all these reasons, they are perfect for school use. These tests were used before and after the intervention programs.

#### 2.4.3. Amos 8-15 Questionnaire

The Amos 8-15 [43] is a set of tests designed for the Italian educational environment. It is meant to evaluate study skills and different motivational factors of students aged 8 to 15. It allows users to recognize students’ weaknesses and strengths in order to launch targeted activities aimed at promoting effective study methods and motivational strategies related to the process of learning.

It includes easy-to-administer tools that help analyze different aspects involved in learning activities, such as approaches to the study, uses of strategies for the study, beliefs about oneself as a student, and accidental attributions about both successful or failed events. This battery test comprises: (i) study approach questionnaire (QAS); (ii) study strategies questionnaire (QS1 e QS2); (iii) convictions questionnaire (QC1I, QC2F, QC3O); (iv) attributions questionnaire (QCA); (v) objective study tests.

The researcher has the possibility to decide to employ all the existing tests or some of them. The QAS and the Objective study tests were chosen by the authors to conduct the research. These tests were carried out at the start and following the experimental trial.

#### 2.4.4. Study Approach Questionnaire (QAS)

It evaluates various variables of the student’s study attitude. The QAS consists of 49 items classified into 7 macro-areas: each area includes 7 items; 5 are positive, and 2 are negative. Anxiety is an exception because it consists of 2 positive and 5 negative items. These macro-areas include: (i) motivation; (ii) organization; (iii) didactic material development; (iv) study flexibility; (v) concentration; (vi) anxiety; (vii) attitude towards school.

The QAS uses a 3-point Likert-type scale, requiring responses ranging from 1 “disagree” to 3 “strongly agree”. This questionnaire needs about 10 to 20 min to be completed (including the instruction and practice phase). If the respondent’s QAS increases, this suggests they are experiencing an inappropriate approach to the study.

#### 2.4.5. Objective Study Tests

The objective study test is an evaluation that allows checking the ability of the student to comprehend and memorize. It provides a text to be studied for 30 min subjectively. Afterward, on completion of a 15 min break, participants perform 3 different tests: Choice of titles wherein students choose 3 most significant titles from a list of 8 titles. For each valid title, it is assigned 1 point.Open questions in which the subject responds to 6 questions concerning the text previously studied. To assess the accuracy of the response, it is assigned from 1 to 3 points.True/False questions ask the student to respond true or false to 12 questions. In addition, 1 point is given for each right response, 0 point is awarded for answers not given, and −1 point for each wrong response.

The sum of the scores obtained determines the total scoring system. This questionnaire required about 75 to 90 min to be completed.

#### 2.4.6. Scholastic Self-Efficacy Scale 

The scholastic self-efficacy scale [44] is a questionnaire thought to evaluate self-beliefs in one’s competence to keep doing well in study and the self-regulation of learning. It is a 19-item questionnaire using a 5-point Likert-type scale. The evaluation requires responses ranging from 1 “not at all capable” to 5 “highly capable”. The evaluation requires about 15 min (comprising the instruction and practice phase). The score ranges from 19 to 95. An increase in self-efficacy is identified with higher scores. This questionnaire was carried out at the start and following the experimental trial.

#### 2.4.7. Physical Activity Intervention

The exercise training intervention for the experimental group was planned in accordance with the social-cognitive theory [45]. It sees self-efficacy perceptions as feelings based on 4 main lines of evidence: past performances, vicarious experiences, verbal persuasion and physiological state [23]. Thus, to enhance the sense of self-efficacy among overweight and non-overweight students, key elements of the intervention program based upon principles of the social cognitive theory were summarized as follows: It was an enjoyable and appealing activity designed specifically to help overweight students to achieve success (i.e., utilizing activities easier to perform and using students’ favorite music);It created occasions for students to positively influence each other (i.e., experience a team mentality);It verbally encouraged students to overcome their limits (i.e., ‘you can do it’);It reduced any kind of anxiety related to exercise, avoiding competition.

Moreover, it was planned as follows: The first part focused on flexibility (3 min), followed by a core part of moderate-to-vigorous aerobic exercise (15 min), and lastly, a cool-down session (2 min) to preserve a safe heart rate. Warm-up parts involved the following exercises: marching in place, walking jacks, knee to chest, heel digs, arm circles, shoulder rolls, knee lifts, butt kicks, lunges, side steps and high knees. Moreover, cool-down parts included static exercises, such as neck stretch, behind-head tricep stretch, standing hip rotation, hamstring stretch, hip flexor stretch, side stretch and butterfly stretch.

Each active break was planned to be performed between and within science classes, during which a program of nutritional education was carried out.

A typical session of an active classroom intervention included:Active breaks between and within learning activities;Learning activities which involve movement;Working at benches, standing desks, on the floor, or in combination to create movement between work areas;Learning outdoors.

The intensity of each training session was monitored through an OMNI scale to respect exertion in the MVPA range of 5 < RPE < 8 and to prevent any differences between training sessions [46].

#### 2.4.8. Nutritional Education Intervention

Nutritional education consisted of a series of interventions that provided visual and verbal information and directions to attendees.

The goals of this intervention were to:Provide a good approach toward healthy food and physical activity and stimulate reasoning for enhanced eating habits and active lifestyles in order to promote well-being for students.Solicit students to think critically about nutrition and healthy lifestyles so that they can make suitable choices about what to eat, excluding junk food.Help the students to critically recognize and analyze appropriate resources of information.

The framework for this nutrition education activity was represented by the *Dietary Guidelines* drawn up by the Italian Association of Dietitians. They comprised information concerning normal-weight maintenance, active lifestyle habits, healthy food and alcohol consumption. Therefore, this program provided information regarding:The recommended food intake as needed to maintain daily nutritional needs;Knowledge about the physiology of the human body;Behavioral practices, including the factors that affect students’ eating and food preparation habits;Different ways to acquire a healthy and active lifestyle;The importance of a healthy diet and physical activity in preventing diseases.

#### 2.4.9. Statistical Analysis

The IBM SPSS version 25.0 (IBM, Armonk, NY, USA) was used to conduct the statistical analyses. Data were introduced as group mean (*M*) values and standard deviations (*SD*). In addition, they were checked for assumptions of normality through the Shapiro–Wilk test and homogeneity of variances by using Levene’s test. To evaluate group differences at baseline, an independent sample *t*-test was used. To explore the impact of the physical activity program on all dependent variables, a two-way ANOVA (group (experimental/control) × time (pre-/post-intervention) with repeated measures on the time dimension was carried out. When ‘Group × Time’ interactions showed significance, a paired *t*-test was performed to outline the significant comparisons. Lastly, to analyze the magnitude of the significant ‘Time × Group’ interaction, a partial eta squared *(η^2^_p_)* value was used. It was interpreted as follows: small (*η^2^_p_* < 0.06), medium (0.06 ≤ *η^2^_p_* < 0.14), and large (*η^2^_p_* ≥ 0.14). Moreover, Cohen’s *d* was used to determine the effect sizes for the pairwise comparisons. It was interpreted as small (0.20 ≤ *d* < 0.50), moderate (0.50 ≤ *d* < 0.79) and large (*d* ≥ 0.80) [47]. Statistical significance was set at *p* < 0.05.

## 3. Results

All participants received the treatment conditions as allocated, and no participants reported injuries during the course of the trial. Students involved in the research did not differ in age, sex, anthropometric characteristics and psychological measures, as well as in socioeconomic status (*p* > 0.05). Data results for all dependent measures are shown in Table 1.

### 3.1. Motor Tests

Using a two-factor repeated measures ANOVA, we found a significant ‘Time × Group’ interaction for all four Motor tests carried out: standing long jump test (F_1,98_ = 23.98, *p* < 0.001, *η^2^_p_* = 0.19, large effect size), Harvard step test (F_1,98_ = 374.01, *p* < 0.001, *η^2^_p_* = 0.79, large effect size), push up tests (F_1,98_ = 363.39, *p* < 0.001, η^2^_p_ = 0.78, large effect size) and sit and reach test (F_1,98_ = 386.83, *p* < 0.001, *η^2^_p_* = 0.79, large effect size). Moreover, post hoc analysis showed that, from pre- to post-test, the experimental group made a meaningful increase in standing long jump test (t = 9.86, *p* < 0.001, d = 1.39 large effect size), Harvard step test (t = 25.78, *p* < 0.001, d = 3.64, large effect size), push up test (t = 20.42, *p* < 0.001, d = 2.88, large effect size) and sit and reach test (t = 19.04, *p* < 0.001, d = 2.69, large effect size). The control group did not report any significant changes (*p* > 0.05).

#### 3.1.1. BMI

A meaningful ‘Time×Group’ interaction was also reached for BMI (F_1,98_ = 59.30, *p* < 0.001, *η^2^_p_* = 0.37, large effect size). Clearly, this case post hoc analysis also showed that the experimental group showed an important decrease in BMI from pre- to post-test (t = 7.97, *p* < 0.001, d = 1.12 large effect size). After the intervention program, the percentage of overweight children decreased from 46% to 28%. The control group did not report any significant changes (*p* > 0.05).

#### 3.1.2. Study Approach Questionnaire QAS

A substantial ‘Time × Group’ interaction was obtained for motivation (F_1,98_ = 162.70, *p* < 0.001, *η^2^_p_* = 0.62, large effect size), organization (F_1,98_ = 186.97, *p* < 0.001, *η^2^_p_* = 0.65, large effect size), study flexibility (F_1,98_ = 328.50, *p* < 0.001, *η^2^_p_* = 0.77, large effect size), concentration (F_1,98_ = 194.89, *p* < 0.001, *η^2^_p_* = 0.66, large effect size) and anxiety (F_1,28_ = 188.68, *p* < 0.001, *η^2^_p_* = 0.65, large effect size). When the post hoc analysis was performed, it showed that the experimental group reached an important increase in motivation (t = 10.20, *p* < 0.001, d = 1.44, large effect size), organization (t = 9.86, *p* < 0.001, d = 1.39, large effect size), study flexibility (t = 13.37, *p* < 0.001, d = 1.89, large effect size) and concentration (t = 10.21, *p* < 0.001, d = 1.44, large effect size). In addition, a significant decrease in anxiety scores was obtained by the EG (t = −12.16, *p* < 0.001, d = 1.72, large effect size). Lastly, no significant “Time × Group” interaction was reached in didactic material development and attitude toward school (*p* > 0.05). The control group did not report any significant changes (*p* > 0.05).

#### 3.1.3. Objective Study Tests

The results of a two-factor repeated measure ANOVA showed meaningful “Time × Group” interaction for objective study tests (F_1,98_ = 220.36, *p* < 0.001, *η^2^_p_* = 0.69, large effect size). Carrying out the post hoc analysis, it was found that EG made a significant increase in the score for objective study tests (t = 10.26, *p* < 0.001, d = 1.45, large effect size). The control group did not report any significant changes (*p* > 0.05).

#### 3.1.4. Scholastic Self-Efficacy

The results of a two-factor repeated measure ANOVA showed a meaningful “Time × Group” interaction for self-efficacy tests (F_1,99_ = 981.36, *p* < 0.001, *η^2^_p_* = 0.90, large effect size). When the post hoc analysis was performed, it showed that the experimental group reached an important increase in self-efficacy tests (t = 40.34, *p* < 0.001, d = 5.70, large effect size). The control group did not report any significant changes (*p* > 0.05).

## 4. Discussion

The aim of this work was to analyze the relationship between a 12-week physical activity classroom break intervention, the perceived sense of self-efficacy and academic achievement among overweight and non-overweight high school students, grounded on the posit that physical activity plays a crucial role as a mediating factor between self-efficacy and the impact of programs on eating behavior and academic achievement. 

In the frame of this work, the findings show that the 12-week active break performed during a program of nutritional education was most effective in promoting the highest academic performance and perceived sense of self-efficacy in overweight and non-overweight high school students. On the contrary, the regular lessons where no practice was performed were less efficient in achieving implications that matched the objectives set out.

The first result of the present study was the significant difference between overweight and non-overweight students with respect to self-efficacy. This difference is most likely attributable to the fact that participants categorized as overweight had poor self-confidence and were considerably less trusting in trying to find opportunities to engage in physical activity and select physically active tasks instead of those sedentary ones than their non-overweight counterparts. Our findings are consistent with previous findings which showed that lower baseline self-efficacy was linked to a higher weight status [48,49,50,51]. Consequently, the sense of perceived self-efficacy was associated with physical activity in overweight individuals [52]. Therefore, the interventions targeting it represent the most effective means to increase physical activity in this population [53]. Specifically, interventions focused on increasing self-efficacy were shown to lead to higher levels of physical activity in overweight adolescents. 

The present findings also showed that as self-efficacy and levels of physical activity increased, the BMI decreased. It was coherent with earlier scientific literature which demonstrated that increased self-efficacy was related to greater weight loss. In fact, it is possible that physical activity programs specifically designed with the aim to enhance students’ self-efficacy could be proven to be effective for intervention outcomes. Otherwise, enhancements in self-efficacy could be the result of weight loss [54,55,56]. Therefore, according to what was discussed, self-efficacy does not work in a direct way on weight, but it affects weight control behaviors. This happens even more if physical activity program is associated with a nutritional education program. It is in accordance with the social cognitive theory whereby a result obtained in a certain domain is the outgrowth of behavior change. At the same time, behavior changes depend on positive experiences and expectations [23,57]. In fact, health behavior theories argued that improvements in self-efficacy over the course of time will result in behavior change. In this frame, self-efficacy seems to be linked with the choice to follow several health behaviors, such as changes in weight and weight-related behaviors [58]. Therefore, it might be useful to promote certain behavioral strategies of self-efficacy (i.e., nutritional education program and physical activity) specially developed to improve the perceived sense of self-efficacy. These strategies will allow encouraging enhancements in eating intake, physical activity, and weight loss [59].

Perhaps, the major result of the present research was proof of the effectiveness of this active break program to enhance the learning skills of participating students. In line with previous research, it seemed that this finding was a consequence of lower anxiety, greater motivation toward learning, improved working memory, concentration, attention skills [60,61,62], and last but not least, greater ability to organize the study and decrease off-task behaviors [63,64]. This result stemmed from the advantages provided by the school-based active break. Several pieces of evidence posit that physical activity may enhance students’ academic performance owing to the fact that they develop faster-responding skills with regard to different cognitive tasks (i.e., executive function skills and on-task behavior) at the end of a training session [65,66,67]. Moreover, their responses become more accurate [68,69]. Indeed, physical activity is a visible mediator of the enhancements in academic achievement since it is able to allow the allocation of neural resources underlying performance on working memory tasks [70,71]. A large body of research has demonstrated that individuals who perform high levels of physical activity show greater problem-solving and decision-making capabilities [72,73]. In addition, this supports the idea that the introduction of an active break in the daily routine may solicit arousal and decrease boredom [74,75]. Dishman and colleagues (2019) [76] argued that boosting physical activity could lead to an improvement in self-efficacy and consequently improved classroom behavior, as well as academic outcomes. Regarding the anxiety-reducing status found at the end of the intervention, it demonstrates that physical activity may decrease symptoms of stress and anxiety-related depression [77], leaving a greater amount of working memory that can be utilized for cognitive tasks. On the contrary, negative feelings stemming from anxiety take up working memory, decreasing the students’ resources required for learning [78]. In fact, it seemed that lower indices of self-concept and higher levels of BMI could interfere with scholastic functioning [79,80]. They could impact school achievement and participation in school physical activity. 

Students with an increase in the sense of perceived self-efficacy and a decrease in weight as a consequence of an active and healthy lifestyle acquired demonstrated greater motivation toward learning, better concentration, reduction of anxiety and higher capacity to organize study tasks relative to sedentary peers [5]. A growing body of scientific research has also revealed a negative relationship between academic achievement and body mass index through the use of self-reported measures of performance [81,82]. Our study offers added value in expanding the scientific evidence thanks to the fact that it reports objective measures of student performance. Moreover, it explores variables that, to the best of our knowledge, had not yet been fully discussed previously, such as the capacity to organize study tasks and be more flexible toward studying. They provide a much better picture of student performance.

Clearly, a meaningful connection between BMI and active lifestyle was reported in this work [83]. Students who enhanced their BMI were also those who reported the best results in the learning ability tests, at the same time improving their health. On the contrary, adolescents that experienced a significant reduction in the sense of perceived self-efficacy and a sedentary lifestyle showed poor academic performance. Earlier research about physical activity among younger demonstrated that lower levels of physical activity appear to be a significant determining factor in the occurrence of negative psychosocial determinants, as well as academic failure [84]. Physical activity is, therefore, an important contributing factor in the development and/or maintenance of self-efficacy and academic achievement.

Lastly, it does not come as a surprise that students who were able to decrease their weight seem to feel better athletically than when they were overweight. This finding is in accordance with the assumption that a lack of physical activity and sedentarism are determinants in the storage of adolescent fatness. Overweight situations in adolescents cause negative long-term implications, given that they have more limited chances to finish school on time, achieve high income, and live a wealthy life as adults than their non-overweight peers [85].

But what is surprising was to find that both overweight and non-overweight students participated in school-based sports teams. There were no differences in sports participation by weight status. This unexpected finding could be a consequence of the “no-cut” policy for sports and physical activity engagement of the selected school. This suggests that organized sports are a viable physical activity option for overweight adolescents.

According to what was discussed, these findings were in line with earlier studies confirming the evidence of a significant association between an active lifestyle, self-efficacy and academic achievement [86,87,88,89,90,91]. Physical activity appears to have several positive effects on academic outcomes. They occur via different direct and indirect physiological, emotional, cognitive and learning processes [92,93,94,95]. For these reasons, schools should provide occasions for students to keep a steady pace of physical activity [96]. Masini et al. (2022) [97] argued that the integration of school-based active breaks during school daily routine is the most profitable way to develop and enhance both cognitive functions and learning performance.

Although this work provides support concerning the positive relationship between physical activity, self-efficacy, weight loss and school outcomes, some limitations of this study warrant consideration. First of all, the present study is restricted by the inclusion of pupils who attended the same school. Therefore, the findings may not be extended to pupils at other schools or with different backgrounds. Moreover, the small sample size (N = 100) spawned the difficulties in engaging participants is another limitation. Lastly, another limitation is the lack of evaluation of the long-term effects of physical activity on cognitive functions. Furthermore, it did not explore a wide age range, and data were collected from a single window of time. Therefore, it is recommended that subsequent research analyzes similar factors on a wider and varied sample which involves pupils in the primary, secondary and high school. Nevertheless, the findings achieved may give significant directions for further studies. Therefore, the strengths of this research were provided by this effective strategy that allow increasing physical fitness and academic performance, as well as psychosocial factors.

## 5. Conclusions

This study suggests that an active lifestyle represents an effective way to maintain psycho-physical health and well-being, particularly among adolescents. In fact, it is of paramount importance that the recommended physical activity levels are achieved and maintained among all age groups. Nonetheless, a lack of physical activity is booming significantly among the youngest generations. This phenomenon is mainly the result of the increase in sedentary behaviors taken during the school day.

According to evidence-based recommendations to promote physical activity in youth, it will be important to integrate more physical exercise into daily school routines and to encourage students whenever they are physically active. Future research should target further approaches to tackle the consequences of too many sedentary activities and to effectively increase physical activity in youth.

Therefore, it is important that the school pays more attention to physical activity during the daily school routine in order to increase self-efficacy and academic performance, as well as to reduce overweight populations.

## Figures and Tables

**Figure 1 nutrients-15-02061-f001:**
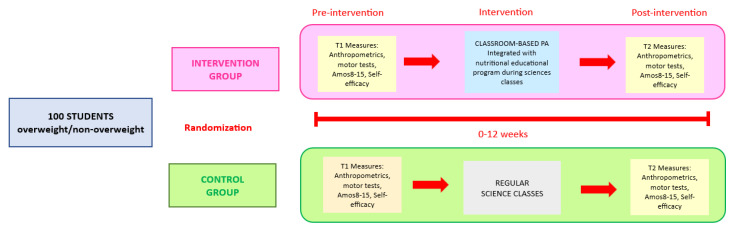
Study design graph.

**Table 1 nutrients-15-02061-t001:** Changes in physical fitness, self-efficacy, study abilities and BMI after a 12-week active break program.

	Experimental Group (*n* = 50)			Control Group (*n* = 50)		
	Baseline	Post-Test	Δ	Baseline	Post-Test	Δ
**Motor Tests**						
Standing long jump test	1.51 (0.06)	1.54 (0.05) †*	0.03 (0.02)	1.49 (0.60)	1.49 (0.06)	0.05 (0.39)
Harvard step test	35.08 (12.98)	39.98 (12.81) †*	4.90 (1.34)	36.72 (13.3)	34.27 (13.41)	−2.45 (2.33)
Push up test	6.36 (1.85)	13.62 (3.50) †*	7.26 (2.51)	5.80 (1.87)	5.64 (1.89)	−0.15 (1.17)
Sit and Reach test	4.82 (2.32)	9.28 (3.10) †*	4.46 (3.18)	4.98 (2.23)	3.56 (2.60)	−1.41 (1.32)
**Amos 8-15—QAS** **Questionnaire**						
Motivation	13.76 (2.09)	15.54 (2.42) †*	1.78 (1.23)	14.24 (2.34)	12.86 (2.10)	−1.38 (1.24)
Organisation	14.96 (1.91)	17.02 (1.96) †*	2.06 (1.47)	15.04 (2.64)	13.72 (2.24)	−1.32 (0.93)
Didactic material development	15.22 (1.50)	14.94 (1.69)	−0.28 (0.75)	14.52 (1.47)	13.52 (1.48) 2.25	−1 (1.06)
Study flexibility	15.66 (1.89)	17.24 (2.02) †*	1.58 (0.83)	15.72 (1.60)	14.18 (1.82)	−1.54 (0.88)
Concentration	16.94 (2.03)	18.26 (1.87) †*	1.32 (0.91)	17.18 (2.08)	15.56 (2.18)	−1.62 (1.17)
Anxiety	17.10 (1.66)	15.02 (1.67) †*	−2.08 (1.20)	15.38 (2.35)	16.50 (2.13)	1.12 (1.11)
Attitude towards school	15.72 (2.44)	15.64 (2.51)	−0.08 (0.56)	16.62 (1.79)	15.12 (1.98)	−1.50 (0.70)
**Amos 8-15—Objective Study Tests**	18.76 (2.97)	22.56 (3.84) †*	3.80 (2.61)	18.98 (2.89)	16.98 (2.99)	−2 (0.88)
**Scholastic Self-efficacy**	46.76 (2.72)	54.12 (2.52)†*	7.36 (1.28)	46.86 (2.80)	44.92 (3.07)	−1.94 (1.66)
**BMI percentile median (QR) ^Ω^**	91.65 (4.01)	88.79 (6.17) †*	−2.84 (2.54)	91.82 (4.03)	91.73 (4.01)	−0.09 (0.36)

Note: values are presented as mean (±SD); Δ: pre- to post-training changes; † Significant ‘Group × Time’ interaction: significant effect of the intervention (*p* < 0.001). * Significantly different from pre-test (*p* < 0.001); **^Ω^** BMI percentile indicates the relative position of the child’s BMI number among children of the same sex and age.

## Data Availability

Not applicable.

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
