# Peer review of "Classroom-Based Physical Activity as a Means to Improve Self-Efficacy and Academic Achievement among Normal-Weight and Overweight Youth"

_nutrients, 2023, doi:10.3390/nu15092061_

Round 1
Reviewer 1 Report
Dear Authors,
I was given a manuscript to check titled: Classroom-based Physical Activity as a Means to Improve Self-efficacy and Academic Achievement among 2 Normal-weight and 3 Overweight Youth
From my point of view, the thesis contains several methodological errors that significantly affect the quality of the article.
I have a number of questions and comments in this regard
The sample consisted of overweight and non-overweight students, while there is no indication of the percentage of overweight students. How were they then divided into experimental and control groups if randomized? Please indicate what percentage of students were overweight and what percentage were in each group. How do these data affect the results presented?
In the Study design it is stated that: "Both interventions consisted of 2 weekly sessions of physical activity lasting 60 minutes performed during the school day". However, in section 2.4.7 Physical Activity Intervention in lines 250 - 252 you state that the students practised 3 minutes of flexibility, 15 minutes of a core portion of moderate-to-vigorous aerobic exercise, and 2 minutes of a cool-down session. This totals 20 minutes of exercise. It is somewhat disturbing that you then break down the warm-up and stretching, while neglecting the main part of the lesson. Although lines 239-249 provide a very interesting presentation of how the intervention was planned, it is only theory
In the Physical Activity Intervention section, you further describe what a typical session of an active classroom intervention would include. I can't quite picture, especially items 2-4 (lines 261-264) how the students performed them at a moderate to high intensity.
In section 2.2 Participants, you state in line 141 that: "The study was conducted from November 2022 to December 2022", but the text below states that it was a 12-week programme. Which of those statements is true?
The results obtained evoke a correction of the data, as all the results came out much too clearly in favour of the experimental group. For although it is scientifically proven that PA correlates with better cognitive function and academic performance, as you also state in the theoretical section, in this version of the article I have my doubts about the experimental agent, i.e. PA. In the present context, my question is whether students in Italy have no TV lessons at secondary school?
Based on the above, among others, please add information about the frequency of physical activity per week at which the authors found significant changes in the parameters monitored (e.g., self-efficacy, academic performance, etc.), which you present in the theoretical framewok.
In the theoretical sectionframework you cite the 2010 WHO guidelines on physical activity, which urges young people from 5 to 17 years to carry out at least 60 minutes of moderate-to-vigorous physical activity 46 (MVPA) per day. Currently available on the web are the WHO recommendations from 2020, I firmly suggest that you update them. Nevertheless, in the context of the above, 2 sessions per week (especially if they were 20min. blocks, not 2x60min. blocks) can't possibly yield such a significant difference between the experimental and control groups in terms of BMI reduction in particular, as I see it.
I suggest that the sentences from lines 163 and 164 be removed.
Please add how many classes of Nutritional Education Intervention the students attended. Also, whether this was in addition to the physical activity classes or inclusive of them?
Please indicate in the results also the average BMI value.
Author Response
Please, see the attached file

Reviewer 2 Report
This topic is of great interest given the rising prevalence of obesity among children and adolescents, along with the increased prevalence of physical inactivity and sedentary behaviours. However, many changes need to be introduced before considering this manuscript for publication. Also, English editing is needed in different parts of the manuscript.
. Abstract: in the Methods section, authors need to clarify if students were recruited from different schools in Italy, or if any specific city/area was focused on. In the Results section, authors should refer to the treatment group as intervention group; also, what type of improvements were seen? this needs to be specified; also, what do authors mean by 'no material change', line 24, for the control group?
. Keywords: more keywords should be added to better reflect the objectives and outcomes of the study.
. Introduction: well written overall. Lines 36-38: schools do not promote intentionally inactivity but there is less focus in many of them on keeping children physically active in different ways; please rewrite this sentence. Lines 44-47: these are not the most recent guidelines by WHO; please update. Also, authors should cite other studies worldwide including in Italy that have tried to boost physical activity in school settings…and if any favourable outcomes were seen; this is important to support the uniqueness and the need for this study. Line 95: abbreviations such as BMI should be explained when first used; also, why do authors expect to see differences in the responses to the physical activity interventions by BMI? Has this been reported before? please clarify.
. Methods: Lines 110-111: what do authors mean by “both the interventions”? Well written overall; suggest to add a figure summarizing the design of the study, and the different tests and outcomes.
. Discussion: Lines 450-455: what do authors mean by “fighting weight”? please rewrite this paragraph or clarify it.
. References: good number and up-to-date. More references could be added as mentioned in the Introduction section, if needed.
Round 2
Reviewer 1 Report
Dear authors,
thank you for respecting my comments. I believe that by doing so, you have improved the quality of your work.
Reviewer 2 Report
Authors have addressed all my comments. English editing is still needed.